# Oxidation of Polyunsaturated Fatty Acids as a Promising Area of Research in Infertility

**DOI:** 10.3390/antiox11051002

**Published:** 2022-05-19

**Authors:** Giulia Collodel, Elena Moretti, Daria Noto, Roberta Corsaro, Cinzia Signorini

**Affiliations:** Department of Molecular and Developmental Medicine, University of Siena, Policlinico Santa Maria alle Scotte, Viale Bracci 16, 53100 Siena, Italy; giulia.collodel@unisi.it (G.C.); noto@student.unisi.it (D.N.); r.corsaro@student.unisi.it (R.C.); cinzia.signorini@unisi.it (C.S.)

**Keywords:** diet, fatty acids, F_2_-isoprostanes, inflammation, male infertility, sperm quality resolvins, oxidative stress

## Abstract

In this review, the role of fatty acids (FA) in human pathological conditions, infertility in particular, was considered. FA and FA-derived metabolites modulate cell membrane composition, membrane lipid microdomains and cell signaling. Moreover, such molecules are involved in cell death, immunological responses and inflammatory processes. Human health and several pathological conditions are specifically associated with both dietary and cell membrane lipid profiles. The role of FA metabolism in human sperm and spermatogenesis has recently been investigated. Cumulative findings indicate F_2_ isoprostanes (oxygenated products from arachidonic acid metabolism) and resolvins (lipid mediators of resolution of inflammation) as promising biomarkers for the evaluation of semen and follicular fluid quality. Advanced knowledge in this field could lead to new scenarios in the treatment of infertility.

## 1. Introduction

Fatty acids (FA) play a fundamental role in cellular mechanisms. With respect to the number of double bonds present within the carbon chain, they are classified as saturated (SFA), monounsaturated (MUFA) and polyunsaturated fatty acids (PUFA). As FA are components of phospholipids in cell membrane structures, also involved in lipid–protein interactions, the length of their carbon chains affects membrane properties, cellular processes and susceptibility to cell death. Thus, the engineering of membrane lipid composition represents a new frontier in nutraceutical and pharmaceutical intervention [1,2]. In particular, SFA and PUFA influence membrane fluidity [3,4] and are also precursors of lipid signaling molecules [5]. Among the n-3 PUFA, eicosapentaenoic (EPA, 20:5n-3) and docosahexaenoic acid (DHA, 22:6n-3) are involved in immunological responses [6], inflammatory processes and, by displacing n-6 PUFA and cholesterol, the modulation of cell membrane composition, membrane lipid microdomains (lipid rafts) [7] and cell signaling [8]. Interestingly, unsaturated FA appear to be involved in lipid ordering and lipid raft stability, also influencing inflammatory effects, given that lipid rafts are platforms for the assembly and function of many signaling pathways [9]. The role of dietary lipids has been debated and supported for gut [10,11] and brain functions [12,13]; moreover, the ability of EPA and DHA to reduce blood pressure and inflammatory processes has been reported [14]. In cells, the membrane FA composition influences the inflammatory response by affecting the production of inflammatory mediators [15]. In fact, the increase in the membrane content of n-3 PUFA (EPA and DHA), at the expense of the arachidonic acid (AA, 20:4n-6) content (an n-6 PUFA), is followed by an increase in the production of eicosanoids and resolvins [16]. Remarkably, it is well established that the inappropriate regulation of inflammation contributes to a range of human diseases. Along these lines, high-fat diets induce high levels of endotoxins, circulating free FA and inflammatory mediators, resulting in metabolic inflammation throughout the organism [17]. Recently, the PUFA biosynthesis pathway has also been invoked in the inflammatory complications of COVID-19 [18].

Phospholipase A_2_ hydrolyzes AA esterified to membrane phospholipids so that its free form is further metabolized by cyclooxygenase and lipoxygenase enzymes to a spectrum of bioactive lipid mediators, including prostanoids and lipoxins, whose receptors are coupled to G proteins and mediate pharmacological effects (prostanoid receptors and lipoxin receptors). The role of AA metabolism in human health and inflammatory-related diseases has been reviewed [19,20]. Human health and several pathological conditions have been shown to be associated with both dietary and cell membrane lipid profiles [21,22,23,24,25] (Figure 1).

## 2. Fatty Acids in Normal and Altered Spermatogenesis

The role of FA metabolism in human sperm and spermatogenesis is a key issue that requires clarification. FA accumulate in testicular cells through passive diffusion and/or protein-facilitated transport, mediated by CD36 glycoprotein expressed in Sertoli cells. In humans and animals, alpha-linolenic acid (ALA, 18:3n-3) and linoleic acid (LA, 18:2n-6) are essential FA that cannot be synthesized and must therefore be provided by the diet. Linolenic acid and ALA metabolites are obtained by an elongation and desaturation process catalysed by enzymes such as elongases 2 (Elovl2) and 5 (Elovl5), Δ6-desaturase (FADS1) and Δ5-desaturase (FADS2). DHA and EPA are derived from ALA metabolism [5].

Sertoli cells are the most relevant cell type in the testis concerning essential FA metabolism. Active conversion of essential FA to docosapentaenoic acid (DPA, 22:5n-6) and DHA was observed in Sertoli cells, which, in rats, show a high expression of Δ5-desaturase and Δ6-desaturase compared to germ cells. Decreased levels of DPA are related to smaller testes [26] and lower fertility, which could be due to poor spermatid maturation. The FA profile of cultured rat Sertoli cells was modified by testosterone, which is involved in the modulation of Δ5 and 6 desaturase activity by PUFA biosynthesis [27].

The key enzymes involved in PUFA metabolism have been described during spermatogenesis and epididymal sperm maturation in stallions [28]. FADS1 was expressed in germinal cells and ELOVL5 in germinal and Leydig cells, whereas FADS2 was not detected. FADS1, FADS2 and ELOVL5 were detected in epididymal vesicles secreted via an apocrine mechanism. Recently, Castellini et al. [29] observed in rabbit testis, using immunofluorescence, that PUFA intermediate metabolites, enzymes and final products showed a different localization in Leydig, Sertoli and germinal cells. Leydig cells showed FADS1, FADS2 and ELOV2; Sertoli cells, FADS2; germ cells, ELOVL5 (meiotic cells) and FADS1/2 (elongated spermatids). Epididymal vesicles were positive for FADS1, ELOVL2 and ELOVL5 as well as DHA, EPA and AA [29].

Human Sertoli cells prefer the metabolic conversion of n-3 FA over n-6 FA, which justifies the high concentration of DHA in sperm. Metabolic diseases, including obesity and type II diabetes mellitus, affect FA availability in Sertoli cells and, consequently, male reproduction [30]. Regarding sperm cells, it is known that the FA profile influences not only sperm motility and vitality but also capacitation, the acrosomal reaction and sperm–oocyte fusion [31]. DHA and palmitic acid (C16:0) are the major PUFA and SFA in human sperm. During spermatogenesis and epididymal maturation, the relative amount of DHA in the sperm plasma membrane increases [32].

Normozoospermic men showed different FA amounts in sperm with respect to seminal plasma [33]. DHA was 6.2 times higher in spermatozoa than in seminal plasma, whereas behenic (C22:0) and oleic (C18:1) acids showed the opposite trend. Palmitic, stearic and oleic acids and DHA were the most prevalent FA in sperm cells (Figure 2). Spermatozoa and seminal plasma FA could be taken as predictors of cryopreservation success [34], since sperm n-3 PUFA, especially DHA, were associated with sperm motility and viability after freezing/thawing. MUFA and SFA in sperm are negatively linked to sperm motility and sperm concentration [34,35,36]. On the other hand, a high concentration of DHA was detected in the spermatozoa of normozoospermic subjects [33,37].

FA profiles have also been investigated in human sperm and semen from individuals with pathological conditions associated with infertility. Collodel et al. [38], in a population of fertile and infertile individuals (idiopathic infertility and varicocele), reported that oleic acid and total MUFA in sperm correlated negatively with sperm concentration, progressive motility, normal morphology, vitality and the fertility index (obtained by sperm TEM analysis mathematically elaborated) and positively with sperm necrosis. The amount of EPA in sperm was positively correlated with necrosis and that of SFA negatively correlated with sperm vitality. Many authors reported that, in sperm from controls, the n-3 FA content increased, and the n-6 FA amount decreased, compared those detected in infertile men [34,38,39].

In cases of asthenozoospermia, high levels of oleic and palmitic acids were measured in seminal plasma [40].

It is well known that PUFA are very susceptible to lipid peroxidation (LPO), which plays a prominent role in many acute and chronic diseases [41]. In cases of varicocele, urogenital infections and idiopathic infertility, pathological conditions that may be associated with inflammatory status, a reduced amount of total n-3 PUFA and DHA was observed [33,37,38,42].

So, sperm membrane FA composition and metabolism are both relevant to sperm maturation processes and fertility. The data suggest that the FA content could represent a good marker of male infertility, and proper dietary integration of FA may be a potential therapy in this field [43].

## 3. Influence of Dietary FA Supplementation on Sperm Quality and Function

Two major issues have been pivotal points of investigation in the field of sperm FA profiles: first, the comparison between FA profiles of fertile and infertile men and second, the effect of dietary FA on sperm FA profiles as well as sperm quality and quantity [44]. A current research interest involves the evaluation of a FA diet in the treatment of male infertility evaluated both in animal models and in humans, where standardization is difficult due to variations in the diet and lifestyle as well as the variability of spermatozoa. There is increasing evidence that dietary fat intake has an impact on semen quality [45,46,47], which is negative in the case of SFA consumption [45].

Nonhuman models suggest that trans fatty acid (TFA)-supplemented diets not only cause decreased spermatogenesis but can, in a dose-dependent manner, reduce the production of testosterone and the testicular mass and promote testicular degeneration [48]. Dietary TFA affect human sperm morphology and oocyte quality by changing the membrane lipid composition which, in turn, leads to impairment in metabolic pathways [49]. In rats, supplementation of a high fat diet with 2.5% olive oil partially counteracts the negative effects on sperm quality by increasing motility, reducing oxidative stress and slightly improving mitochondrial efficiency [50]. In mice fed a high-fat diet (Dio Rodent Purified diet with 60% energy from fat, Labdiet) for over 2 months, a decrease in DHA in the testis was associated with impairment of the sperm acrosome reaction and fertility [51].

Different animal studies have shown that dietary n-3 FA are incorporated into spermatozoa, but their effect on semen quality is inconsistent [52]. Some years ago, Lewis et al. [53] observed that spermatozoa with a high PUFA content were susceptible to LPO that could further lead to DNA damage; however, Kelley et al. [54] reported that n-3 PUFA decreased LPO. Recently, in rabbits, supplemental dietary n-3 PUFA (one diet enriched with 10% extruded flaxseed and another with 3.5% fish oil for 110 days) improved sperm motility traits and resulted in an enrichment of membrane FA in the sperm and testes, even if such an increased amount of PUFA negatively affected sperm oxidative status [55]. In the same animal model, diets modulated the expression pattern of Toll-like receptor 4 and proinflammatory cytokines on the hypothalamic-gonadal axis and reproductive organs [56].

In human semen, n-3 PUFA supplementation resulted in higher antioxidant activity; enhanced sperm concentration, motility and morphology [47,57,58] and reduced sperm DNA fragmentation [59]. Human testicular volume was positively related to the intake of n-3 PUFA and negatively related to the intake of n-6 PUFA and TFA [60]. A considerable number of infertile men with idiopathic oligoasthenoteratozoospermia might benefit from n-3 FA (DHA + EPA) administration (1.84 g/d for 32 weeks), resulting in higher antioxidant activity in human seminal fluid, which can enhance sperm count, motility and morphology [48,61]. Alteration of the content and ratio of n-6 and n-3 FA in the diet has been found to influence eicosanoid synthesis and metabolism and affect fertilising ability in males [62]. An increase in the n-3/n-6 PUFA dietary ratio is valuable to sustain the reproductive capacity of male turkeys, especially as they age [63]. Dietary DHA, more efficiently than AA, restored fertility, sperm count and spermiogenesis in DPA and DHA-deficient Δ-6 desaturase-null mice [64]. Recently, a meta-analysis (randomized controlled trials) indicated that supplementing infertile men with n-3 FA (DHA or EPA treatments either alone or in combination with other micronutrients) resulted in significant enhancement of sperm motility concomitant with an increased concentration of seminal DHA [65] and an improvement in the semen quality of infertile and fertile men from couples seeking fertility treatment [66].

At present, we believe that nutrition can both negatively and positively affect semen quality. In this context, a meta-analysis of 16 randomized controlled trials showed that semen parameters improve after n-3 supplementation and decrease with a diet rich in SFA and TFA. These data confirmed the relevant role of a controlled FA diet in male fertility [67].

## 4. Current New Indices of Male Infertility Involving PUFA Oxidation

### 4.1. F_2_-Isoprostanes

Oxidative stress (OS) is caused by an imbalance between the production of reactive oxygen species (ROS) and their quenching by antioxidant compounds that act as defence mechanisms [68]. It is associated with the pathophysiology of various diseases related to male infertility, such as varicocele, leukocytospermia and urogenital infections [69], even though ROS, within a physiological range, are necessary for sperm motility, capacitation, the acrosomal reaction and oocyte interaction [70]. Spermatozoa are particularly susceptible to damage by ROS as their plasma membrane is rich in PUFA, acquired during testicular and epididymal maturation. In spermatozoa, LPO leads to cellular dysfunction due to loss of the membrane fluidity and integrity necessary for successful sperm–oocyte fusion [71]. Primary products resulting from this mechanism include malondialdehyde (MDA), 4-hydroxynonenal (4-HNE) and acrolein [69,72].

MDA is an essential and widespread biomarker for the analysis and monitoring of PUFA peroxidation [73]. In addition, 4-HNE and acrolein form adducts with several sperm proteins, such as axonemal proteins, compromising sperm motility and, in general, sperm function. Furthermore, 4-HNE can bind to mitochondrial proteins in human sperm, triggering the loss of electrons and ROS formation; the resulting OS causes activation of the intrinsic apoptotic cascade, loss of MMP, DNA damage and, finally, cell death [71].

As secondary products of LPO, a series of prostaglandin (PG)-like molecules called isoprostanes (IsoPs) and monocyclic and serial cyclic peroxides have been identified. IsoPs have been detected in mammalian plasma, urine, cerebrospinal fluid, sputum, saliva, brain tissues, atherosclerotic plaques and gastric mucosa [74], and are considered a ‘gold standard’ biomarker of endogenous LPO [75].

Our group studied the role of IsoPs in semen and suggested that these molecules could represent new indices for the evaluation of semen quality and the pathogenesis of infertility, indicating possible personalized therapeutic approaches (Figure 3). They are initially formed in situ on phospholipids, as an esterified form, and then released as free IsoPs into the circulation by the action of phospholipase A_2_; this process does not require the action of the cyclooxygenase enzyme. These products include F_2_-IsoPs from AA, F_2_-diomo-isoprostanes (F_2_-diomo-IsoPs) from adrenic acid (AdA) and F_4_-neuroprostanes (F_4_-NeuroPs) from DHA, and they are all considered LPO indices. Among the IsoP groups, F_2_-IsoPs are considered a reliable biomarker of endogenous LPO as they are ubiquitous in the organism and chemically stable in biological fluids [76]. In humans, F_2_-IsoPs are commonly measured in plasma and urine [77] and, as of recently, represent a valid marker of oxidative damage in semen [78].

The production mechanism of F_2_-IsoPs includes various steps, generating four F_2_-IsoP regioisomers, each composed of eight racemic diastereomers, for a total of 64 compounds. The four classes of regioisomers (5-, 8-, 12- and 15-F_2_-IsoPs) are termed according to the number of carbon atoms on which the hydroxyl group of the side chain is attached [76]. Among these, the most studied is 15-F_2t_-IsoP, also known as 8-iso-prostaglandin F_2α_ (8-iso-PGF_2α_ or iPF_2α_-III). Eight-iso-PGF_2α_ can be generated during inflammation by prostaglandin endoperoxide synthase, in both the free and the phospholipid-esterified form. The latter, which is the most abundant, is not a substrate for prostaglandin endoperoxide synthase [79]. Therefore, free 8-iso-PGF_2α_ represents an efficient tool for identifying LPO events in biological fluid. The level of 8-iso-PGF_2α_ generation in normal condition was reported by van’t Erve et al. [79] in a meta-analysis of published data. Urine has the highest average concentration (1200 ± 600 pg/mL); on average, ~100-fold less is detected in plasma (45.1 ± 18.4 pg/mL) and exhaled breath condensate (30.9 ± 17.2 pg/mL). The study of IsoPs in the male infertility field is growing with the purpose of identifying new indices for detecting the presence and progression of oxidative stress/inflammation, as well as for evaluating the efficacy of treatments; these potential biomarkers should be stable and measurable by non-invasive methods [80]. In seminal plasma, F_2_-IsoPs can be detected both in a free form and in cells esterified to membranes [78]. An increased amount of F_2_-IsoPs was quantified in the semen of men with varicocele [81,82,83] and urogenital infections [83], pathologies generally associated with high ROS levels [84] and inflammation [69].

Among the different aims was the definition of a normal range for F_2_-IsoP levels in relation to their possible clinical use in discriminating the conditions of male infertility associated with the presence of inflammation. Moretti et al. [83] identified a concentration threshold (29.96 ng/mL) of F_2_-IsoPs able to discriminate a physiological status of human semen from pathological conditions related to inflammation. To define this cut-off, seminal levels of F_2_-IsoPs were assessed in 192 patients grouped on the basis of clinical diagnosis: idiopathic infertility (no. 41), urogenital infection (no. 52), varicocele (no. 54) and fertile men (no. 45). The concentration threshold, identified by the ROC curve, was able to discriminate fertile from infertile samples; in particular, 44 out of 45 fertile men were under the defined cut-off (29.96 ng/mL).

From another point of view, F_2_-IsoPs could be a useful marker for evaluating in vitro LPO resulting from gamete handling and in cryopreservation procedures, as well as the efficacy of in vitro antioxidant supplementation. Indeed, sperm exposure to ROS is exacerbated during common laboratory practices [68,84]. For this purpose, Noto et al. [85] demonstrated in vitro that F_2_-IsoP levels increased when human sperm were treated with an oxidant agent and decreased when an antioxidant compound was added. Chlorogenic acid (100 μM) showed a protective effect against the alterations detected in the samples treated with H_2_O_2_ (100 μM) for 1 h. High F_2_-IsoP levels were associated with reduced percentages of sperm with double-stranded DNA and high mitochondrial membrane potential. To confirm the effectiveness of F_2_-IsoPs, the amount of MDA, widely used to evaluate oxidative insult [86], was measured by HPLC. MDA levels showed an analogous trend of F_2_-IsoPs, validating the effectiveness of F_2_-IsoPs as an index of in vitro LPO.

When evaluating the role of F_2_-IsoPs as a marker of LPO in seminal fluid, another aspect could be considered. It has been reported that the plasma concentration of F_2_-IsoPs could be modulated by the administration of defined diets [87,88]. In humans, diet is difficult to standardise, and in the study of reproduction, isolated cells or tissues, when used as a model, cannot describe the steps of maturation of spermatogenesis; consequently, they cannot reflect the effect of diet during this process. Thus, an in vitro approach may be limiting.

Recently, the seminal F_2_-IsoP amount was assessed on rabbit bucks used as an animal model after different dietary plans described above. Their diet was enriched with flaxseed, which has a very high ALA content, whereas a fish oil diet directly supplies ALA derivatives (EPA, DPA and DHA). F_2_-IsoPs were reduced in the semen and blood of rabbits fed both n-3 PUFA dietary sources. Considering that F_2_-IsoPs are produced by AA, which is poorly represented in these diets, the data suggested that cell membranes were enriched in n-3 PUFA obtained by dietary intake [55].

Moreover, the beneficial effect of these FA, indicated by the reduced levels of F_2_-IsoPs, was confirmed by the improvement of rabbit sperm motility and track speed [55]. In this model, F_2_-IsoP levels were higher in both epididymides and testes of the controls than in those of the n-3 PUFA dietary groups [89]. The main regulators of FA metabolism are the peroxisome proliferator-activated receptors (PPARs), transcription factors activated by metabolic ligands. n-3 PUFA diets, by reducing the levels of F_2_-IsoP, proinflammatory molecules linked to LPO, may also influence PPARγ expression and play a role in supporting sperm maturation [89]. For this reason, the evaluation of F_2_-IsoP levels in both epididymides and testes may indicate decreased inflammation after n-3 PUFA-enriched diets. These data agree with the observation that n-3 PUFA could modulate FA composition in cell membrane phospholipids, leading to a decrease in eicosanoids derived from AA, such as prostaglandin E_2_ or leukotriene B_4_ [90].

An interesting observation on the role of F_2_-IsoPs in seminal fluid was provided by a clinical study carried out on patients undergoing assisted reproduction techniques [91].

F_2_-IsoPs were measured in semen samples of 49 infertile men. Semen samples that produced high-quality embryos showed a higher percentage of sperm with double-stranded DNA and increased F_2_-IsoP levels, compared to those that generated low-quality embryos. The amounts of F_2_-IsoPs were slightly increased but were still below the identified cut-off point (29.96 ng/mL) [83] in the semen of men who provided good-quality embryos, compared to the low-quality embryo group. Therefore, the relationship between a mild increase in seminal F_2_-IsoP levels, DNA integrity and high embryo quality suggests that low F_2_-IsoP levels in human semen do not indicate the presence of oxidative stress; rather, they represent a physiological condition (Figure 3).

Concerning the physiological role of IsoPs, Signorini et al. [92] reported that definite concentrations of F_4_-NeuroPs, derived from the oxidative metabolism of DHA, were able to stimulate sperm capacitation.

The main limitation in oxidative stress assessment, due to analytical problems of specificity and sensitivity, resides in the selection of a marker with the highest possible accuracy.

### 4.2. What Is the Role of Resolvins?

In several chronic human pathologies, among which cardiovascular disease, neurodegenerative diseases and respiratory diseases can be counted, derangements of the inflammatory responses are involved. Thus, if acute inflammation is a physiological defensive process, the shift from an acute to a chronic (unresolved) inflammatory profile is linked to the development of several diseases. Therefore, these conditions are grouped in the classification of chronic inflammatory diseases, in which cellular and tissue matrix degenerative events and, finally, loss of functionality occur. Thus, the growing evidence in support of the relevance of chronic inflammation in several diseases leads to speculation on the so-called ‘inflammatory theory of disease’ [93,94]. In understanding the imbalance of the inflammatory processes, the evaluation of inflammation resolution pathways is becoming important. The inflammatory resolution phase is now widely recognized as a biosynthetically active process, governed by a superfamily of endogenous chemical mediators, specialized pro-resolution lipid mediators (SPMs) that induce resolution of inflammatory responses. SPMs comprise a class of bioactive lipids and cell signalling molecules (referred to above as pro-resolving inflammatory mediators) that act to re-absorb inflammatory exudate, stop inflammation and remodel tissues. Tissue remodelling is a typical event in the resolution of inflammation that aims to repair lesions caused by etiological agents and to which the activity of inflammatory cells contributes [95].

The resolution of inflammation is an active process driven by unique signalling molecules. An important group of these SPMs is derived from PUFA (AA, EPA and DHA), which are released during the inflammatory process. In particular, SPMs include (i) n-6 arachidonic acid-derived lipoxins; (ii) n-3 EPA and DHA-derived resolvins, protectins and maresins; (iii) cysteinyl-SPMs and (iv) n-3 docosapentaenoic acid (DPA)-derived SPMs. Thus, lipid mediators have crucial roles in both the initiation of inflammation and its timely resolution. Along the pathway from the initiation to the resolution of inflammation, temporal lipid mediator class switching occurs. Thus, a correct balance and availability of AA, DHA, EPA and other n-3 PUFA may provide tissue protection [96]. The different involvement of PUFA in inflammatory and resolution processes of inflammation, through the different lipid mediators of which they can be precursors, has already been indicated (Figure 4).

Resolvins are classified as E-series and D-series resolvins, whose PUFA substrates are EPA and DHA, respectively. T-series resolvins (RvTs) and RvD n-3 DPA, are classified as n-3 DPA-derived resolvins [96].

In several pathological conditions, SPM biosynthesis impairment has been hypothesized as a disease-causing agent. Moreover, the metabolic pathway and chemical structure of SPMs has been validated in different human tissues [96]. Detailed clinical data on the effect of supplementation with n-3 or marine oil on the production of SPMs in different biological samples (tissues and fluids) have been reported [96].

In male infertility, the involvement of the inflammatory process is known and associated with impaired spermatogenesis [97]. Increased levels of inflammatory cytokines, leukocyte counts and oxidative stress are highly detrimental to sperm quality, thus compromising male fertility [67,98,99]. In particular, our personal contribution indicated that resolvin D1 (RvD1), a lipid mediator enzymatically derived from DHA able to elicit anti-inflammatory and pro-resolving activities [100], was higher in the semen of patients with leukocytospermia, varicocele and idiopathic infertility, compared to that of fertile men. It was positively correlated with LPO (AA peroxidation, F_2_-IsoP production) and reduced sperm quality; in addition, RvD1showed a relationship with membrane lipid composition, seminal ferritin and F_2_-IsoP levels [101]. Therefore, RvD1 appears to be a promising biological indicator to be included in a panel of seminal inflammatory markers for a more accurate diagnosis of inflammatory male infertility and a better definition of personalized treatments.

The role of SPMs in pregnancy and fetal conditions has also been reported [102,103]. In the case of polycystic ovary syndrome, a pro-inflammatory state was detected, and a high pro-inflammatory mediator/SPM ratio was documented in serum [104]. Interestingly, resolvin E1 (RvE1) has been proposed as a robust biomarker for oocyte selection. Actually, RvE1 seems to play a role in improving oocyte quality in humans by increasing cell viability and proliferation [105]. Thus, interest in resolvins in biomedical applications in the female reproductive system is also increasing.

In in vitro and in vivo studies related to chorioamnionitis, the anti-inflammatory role of RvD1, which modulates the PPARγ/NF-κB pathway, was confirmed [106]. In addition, pre-eclampsia is associated with deficient levels of RvD1 and maresin 1, the last one being a further compound involved in the resolution of inflammation. Interestingly, reduced levels of SPMs (i.e., RvD1 and maresin 1) were concomitant with the overproduction of the proinflammatory mediator leukotriene B4 [107]. More details on the biological relevance of resolvins to male and female infertility and to pregnancy complications are displayed in Figure 4.

Thus, the evaluation of resolvins and pro-resolvin-related metabolic pathways could be a promising field in which to investigate the role of inflammatory status, the persistence of a pro-inflammatory status and the occurrence of clinical complications caused by a persistent inflammatory status in infertility conditions. In parallel, investigations into the modulation of the inflammatory pathway could represent a tool to identify the optimal conditions for a successful in vitro fertilization outcome.

## 5. Conclusions

In this review, we have focused on the role of FA and their oxidation in male fertility. Given their role, it has been suggested that isoprostanes derived from membrane LPO may represent a new marker of oxidative stress, offering new targeted therapeutic possibilities involving dietary regimes.

The relevant evidence for FA and lipid mediators, as sound biomarkers of reproductive efficiency, suggests that great attention should be paid to lipid molecules in the optimization of assisted fertilization outcomes.

## Figures and Tables

**Figure 1 antioxidants-11-01002-f001:**
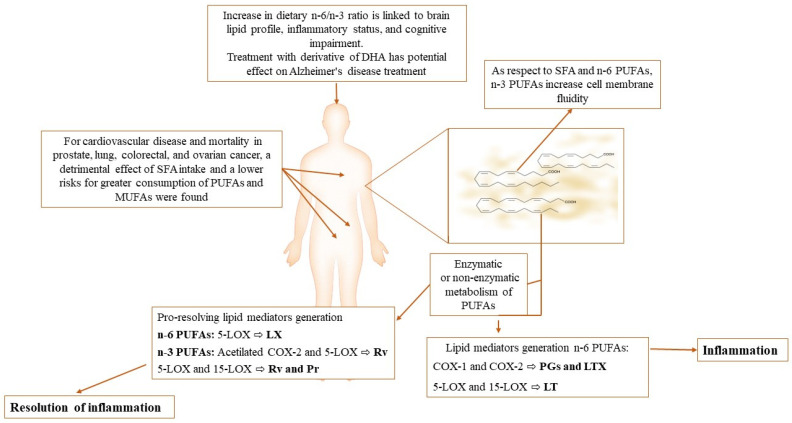
Role of polyunsaturated fatty acids (PUFA) in the modulation of the inflammatory process and in various pathological conditions [22,23,24]. The biophysical features of the cell membrane are conditioned by the membrane fatty acid (FA) composition [3,4]. PUFA metabolism is involved in the release of pro-inflammatory and pro-resolution mediators. A different contribution is attributed to n-3 PUFA (mainly pro-inflammatory mediators) and n-6 PUFA (main pro-resolving inflammatory mediators) [15]. The n-6/n-3 PUFA ratio is related to both the onset and the progression of several diseases [22,23]. References displayed in brackets refer to the references list. Legend: COX, cyclooxygenase; LOX, lipoxygenase; LX, lipoxins; Rv, resolvins; Pr, protectins.

**Figure 2 antioxidants-11-01002-f002:**
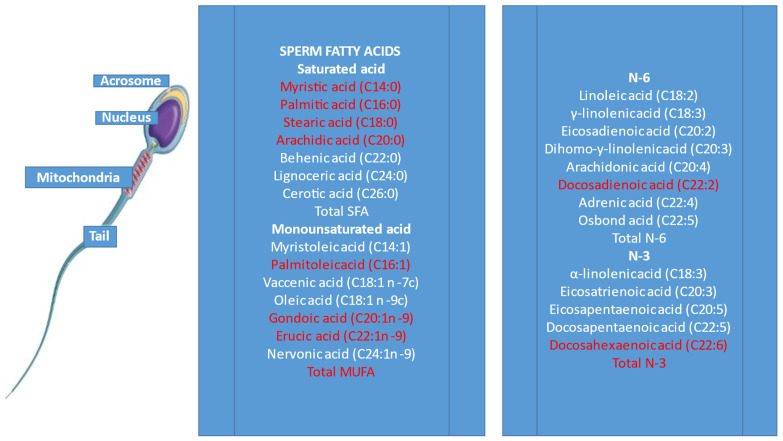
Fatty acids (FA) in human sperm. FA amounts differed in normozoospermic men (white) compared to men with altered semen parameters [33,38].

**Figure 3 antioxidants-11-01002-f003:**
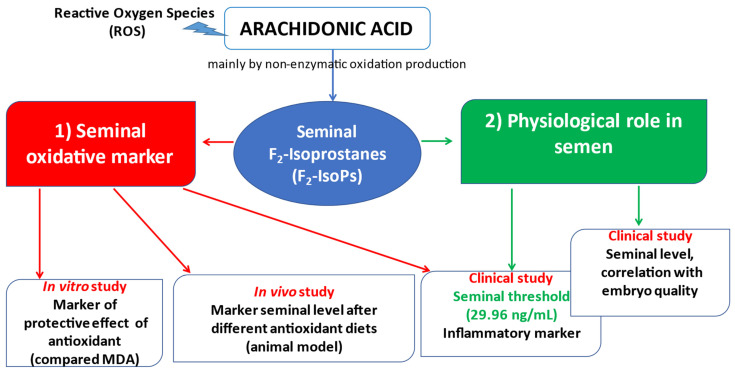
Diagram showing the seminal role of F_2_-Isoprostanes (F_2_-IsoPs).

**Figure 4 antioxidants-11-01002-f004:**
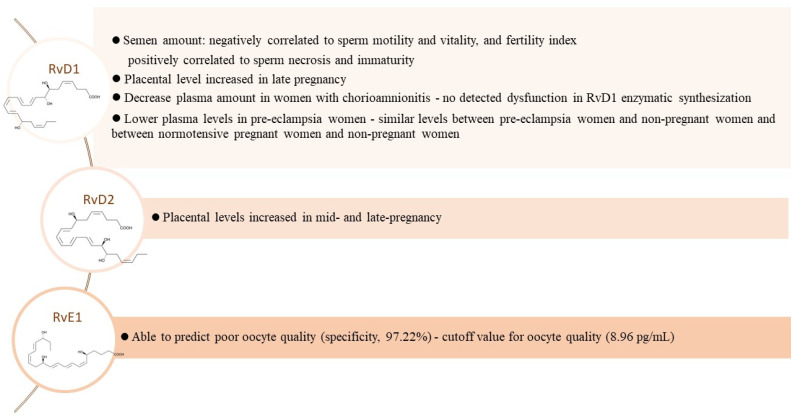
Main involvement of resolvins D (RvD1 [101,103,106,107], RvD2 [103] and RvE1 [104]) in conditions linked to fertility. Resolvins D and E are DHA- and EPA-derived resolvins, respectively. References displayed in brackets refer to the references list. Legend: RvD1, resolvin D1, RvD2, resolvin D2; RvE1, resolvin E1.

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
