# Peer review of "Oxidation of Polyunsaturated Fatty Acids as a Promising Area of Research in Infertility"

_antioxidants, 2022, doi:10.3390/antiox11051002_

Round 1

Reviewer 1 Report

The review manuscript is interesting; however, it is recommended to attend to the following:

Line 8: delete repeated words (correspondence)

Line 9: remove bold text formatting for word (In)

Line 9-20: the size and color of the text is visually different in this section, so it is necessary to standardize the text format

Line 27: change SAFA to SFA, commonly SFA is used to abbreviate saturated fatty acid. Modify through the document

Line 31: remove space between the comma and the reference number [1, 2], according to the Microsoft Word template, it must follow the following format: e.g., [1] or [2,3], or [4–6]. Modify through the document

Line 36: modify…microdomains (lipids rafts) [7], and….

Line 37: unsaturated FA (UFA) ?

Line 49: modify…[17]. Recently, …

Line 58: According to the Microsoft Word template, it must follow the following format: e.g., [21–25].

Line 68: in this line it inserts an abbreviation for the word resolvins, however, in the document text where this word first appears it is not abbreviating it; which is correct?

Line 84: delete space (∆5)

Line 88,92: did you mean FADS2 instead 2?

Line 96,92: the reference was missed

Line 103,115,125,149: SFA?

Line 106: the resolution of the figure should be improved. In some names of the fatty acids listed in the columns of the figures, the words are joined, it is necessary to insert a space (e.g. linolenicacid). Also, the reference number must be placed in the figure, and not the year as it is included. In addition, include the nomenclature of the fatty acids described in the figure, as in line 103, e.g. palmitic acid (C16:0)

Line 116:… [34–36]

Line 130,131: the reference was missed

Line 144-147: was the reference missed?

Line 149: … [44–46]

Line 155: indicate conditions of use of olive oil

Line 157: When a high-fat diet is indicated in this case, to what amount does it refer? and for how long?

Line 163,166: What addition levels were used and for how long?

Line 166: What were the doses of supplementation used, and for how long were they supplemented?

Line 173,174: indicate levels and time of supplementation

Line 181-183: according to the work consulted (meta-analysis), is it possible to indicate the levels and times of supplementation?

Line 140-190: could include a table, indicating the source of FA or type of food, and the effect of supplementation of these through diet on the quality and function of sperm?

Line 215-217: was the reference missed?

Line 225: indicate the range in which the levels of F2-IsoPs are considered normal and/or low-high in plasma and urine, and semen

Line 241: … [80–82]

Line 253,263,270,397: change in vitro by in vitro

Line 258: What antioxidant was used, at what levels and for how long was the supplementation carried out?

Line 271-275: was missed the reference?

Line 272: indicate diet conditions

Line 297:… ng/mL) [82]

Line 303: delete ( - )

Line 321: delete (--)

Line 323: delete ( - )

Line 333-342: was the reference missed?

Line 334,352: PUFAs or PUFA like in the abstract?

Line 371: correlated with LPO (AA …

Line 398: the font included in figure 4 is different from the one used in the text of the manuscript; some words in the figure are superimposed or crossed out; ml or mL?

Line 404: remove bullet in conclusion numbering

Line 434: according to the Microsoft Word template format, the volume of each reference must be written in italics; also use (–) instead (-) of y to separate the number of pages. Some references appear separated by lines and others without space, what is the correct format?. Modify on all references

Line 440: scientific names should be written in cursive text format

Line 458,459,463,490,492,532,544,576,57,611,633,649,etc: Not all words should be capitalized in the title of the cited manuscript.

Line 466: in some references the doi is included and in others it is not, what is correct?

Line 627: insert space ….25 years

Reviewer 2 Report

The manuscript “Polyunsaturated fatty acid oxidation as a promising field in infertility investigations” by Collodel G. with co-authors provides some new and interesting information on the role of fatty acids and their oxidation in male fertility with a focus on F2 isoprostanes (which are oxygenated products from arachidonic acid metabolism) and resolvins (which are lipid mediators of inflammation), which in general are promising biomarkers for assessing the quality of semen and follicular fluid. The topic is important and worthwhile.

However, after reading the entire manuscript, I believe that it requires a lot more work before it being considered acceptable for publication. I found many stylistic and grammatical errors. The abundance of stylistic errors makes it difficult to understand the meaning. English should be improved.

I suggest to modify the title to something like this: "Oxidation of polyunsaturated fatty acids as a promising area of research in infertility"

No dot after title.

Why do the authors divide each sentence into paragraph?

The authors should hold uniform FA name and use the formula at first mention, for example, palmitic acid (16:0) etc.

L.78 “The Sertoli cell” to be replaced by Sertoli cells.

L.79 Incorrect: “Sertoli cells actively convert the essential FA”, because enzymes convert fatty acids in cells. To be corrected such as: Active conversion of the essential FA was observed In Sertoli cells.

L.97 What is “sperm DHA”? Do you mean DHA in sperm? And the future: “Sperm EPA amount” L. 124. It will be better: amount of EPA in sperm.

In Figure 2 alpha-linolenic acid has come into the n-6 fatty acid family.

The words in situ, in vitro must be in Latin.

L.310-322 This paragraph is off topic.

L.324 “A role for resolvins?” May be: "What is the role of resolvins?" Or "Role for resolvins"

L.326 Metabolic syndrome has nothing to do with chronic human pathologies.

There are the problems with Figure 4. The formulas are too small to make out; and legends to the formulas are unreadable.

The list of references should be adjusted to meet the requirements of the rules "Antioxidants".

Round 2

Reviewer 2 Report

The authors followed my recommendations and corrected the manuscript.